# Overcoming *mcr-1* mediated colistin resistance with colistin in combination with other antibiotics

Craig R. MacNair[1], Jonathan M. Stokes[1], Lindsey A. Carfrae[1], Aline A. Fiebig-Comyn[1], Brian K. Coombes[1], Michael R. Mulvey[2] & Eric D. Brown[1]

Plasmid-borne colistin resistance mediated by *mcr-1* may contribute to the dissemination of pan-resistant Gram-negative bacteria. Here, we show that *mcr-1* confers resistance to colistin-induced lysis and bacterial cell death, but provides minimal protection from the ability of colistin to disrupt the Gram-negative outer membrane. Indeed, for colistin-resistant strains of Enterobacteriaceae expressing plasmid-borne *mcr-1*, clinically relevant concentrations of colistin potentiate the action of antibiotics that, by themselves, are not active against Gram-negative bacteria. The result is that several antibiotics, in combination with colistin, display growth-inhibition at levels below their corresponding clinical breakpoints. Furthermore, colistin and clarithromycin combination therapy displays efficacy against *mcr-1*-positive *Klebsiella pneumoniae* in murine thigh and bacteremia infection models at clinically relevant doses. Altogether, these data suggest that the use of colistin in combination with antibiotics that are typically active against Gram-positive bacteria poses a viable therapeutic alternative for highly drug-resistant Gram-negative pathogens expressing *mcr-1*.

[1] Michael G. DeGroote Institute for Infectious Disease Research, Department of Biochemistry and Biomedical Sciences, McMaster University, Hamilton, ON L8N 3ZS, Canada. [2] National Microbiology Laboratory, Public Health Agency of Canada, Winnipeg, MB R3E 3R2, Canada. Correspondence and requests for materials should be addressed to E.D.B. (email: ebrown@mcmaster.ca)

The widespread emergence of carbapenem-resistant Enterobacteriaceae has significantly increased dependence on the cationic peptide colistin, widely regarded as an antibiotic of last resort. Colistin acts by associating with the anionic lipopolysaccharide (LPS) component of the Gram-negative outer membrane, causing membrane destabilization that leads to cell envelope permeability, leakage of cellular contents, and ultimately lytic cell death[1,2]. The recent increase in the use of colistin in clinical practice, accompanied by its unbridled use in agriculture, have contributed to the rapid dissemination of resistance.

Colistin resistance is predominantly achieved through a reduction of the electrostatic attraction between colistin and the Gram-negative outer membrane. This is typically facilitated by the addition of cationic phosphoethanolamine (pEtN) and/or 4-amino-4-deoxy-L-arabinose (L-Ara4N) moieties to phosphate groups on the lipid A component of LPS, which reduces the net anionic charge of the cell surface[3]. Formerly, these LPS modifications were thought to be solely the result of chromosomal mutations that constitutively activate the two-component regulatory systems PhoP-PhoQ and PmrA-PmrB[4,5]. The inability of this form of chromosomal colistin resistance to rapidly spread through bacterial populations via horizontal gene transfer has limited its clinical impact to localized and controllable outbreaks[6]. However, the recent discovery of *Escherichia coli* harboring plasmid-borne colistin resistance via the *mcr-1* gene provides a mechanism for rapid dissemination[7].

Encoding a pEtN transferase, *mcr-1* confers colistin resistance through the addition of pEtN to the lipid A component of LPS. Since its discovery in late 2015, bacteria harboring *mcr-1* have been detected in environmental and hospital isolates worldwide[7,8]. Particularly concerning are those multidrug-resistant strains containing *mcr-1* alongside extended-spectrum β-lactamase and carbapenemase resistance genes[9]. Indeed, the spread of *mcr-1* threatens to decrease the therapeutic utility of colistin from an already shrinking antibiotic arsenal.

With no reprieve from our therapeutic reliance on colistin in the current antibiotic pipeline, bridging the gap between widespread colistin resistance and the development of new antibiotics will require creative use of available treatment options. To this end, the documented ability of colistin to potentiate a variety of antibiotics against Gram-negative pathogens provides an attractive therapeutic opportunity. Synergy with colistin has been explored for a range of antimicrobial agents, most commonly rifampicin[10,11] and carbapenems[12], but also macrolides[13], minocycline[14], tigecycline[15], and glycopeptides[16]. Remarkably, colistin can impact the surface integrity of intrinsically colistin-resistant bacteria[17,18] and antibiotic potentiation is maintained within many pathogens expressing chromosomally mediated resistance to colistin monotherapy[19–21].

Unlike the monogenetic nature of *mcr-1*, chromosomal colistin resistance is mediated through mutations in two-component regulatory systems PhoPQ and PmrAB that activate numerous genes involved in LPS modifications (i.e., *pmrHFIJKLM*) and a plethora of other cellular processes. For example, the PhoPQ system controls the expression of ~ 3% of the *Salmonella* genome[22]. Indeed, in PhoPQ-mediated colistin-resistant *Klebsiella pneumoniae*, additional components of the PhoPQ regulon such as the outer membrane lipoprotein *slyB* and the magnesium-importing ATPase *mgtA* are upregulated, potentially further contributing to resistance beyond pEtN/L-Ara4N modification of lipid A[23]. Comparatively, *mcr-1* reduces colistin efficacy through expression of a single pEtN transferase and does not confer co-resistance to other cationic antimicrobial peptides, as observed in chromosomally regulated colistin resistance[24]. Additionally, *mcr-1*-expressing strains are unaffected by the outer membrane disruption and potentiation of antibiotics by polymyxin B

nonapeptide (PMBN)[25]. Encouragingly, the combination of amikacin and colistin has shown promising combinatorial synergy in carbapenem-resistant *E. coli* carrying *mcr-1*[26]. While significant attention has been directed towards exploring combination treatment options to combat chromosomally mediated colistin resistance, the unique monogenetic nature of *mcr-1* largely differentiates this form of colistin resistance. As such, data generated from chromosomal colistin resistance studies may not accurately predict *mcr-1* susceptibility to antibiotic potentiation with colistin. Therefore, it is essential to thoroughly investigate colistin-based combination therapies against pathogens expressing *mcr-1*.

Here, we screened a collection of Enterobacteriaceae expressing the *mcr-1* gene against a range of antibiotics, representing all major drug classes, for a reduction in minimum inhibitory concentration (MIC) in the presence of colistin. Large, hydrophobic antibiotics conventionally active against Gram-positive bacteria such as rifampins and macrolides demonstrated the greatest decrease in MIC in combination with colistin. Investigating the mechanism underlying this antibiotic potentiation, we observed that *mcr-1* provides a high degree of resistance to the bactericidal and lytic activity of colistin, but confers minimal protection to its outer membrane perturbation. Exploiting this susceptibility through colistin combination treatment demonstrated encouraging efficacy in two mouse models of *mcr-1*-positive *Klebsiella pneumoniae* infection. Additionally, unlike traditional monotherapy antibiotic treatments, resistance to colistin combination therapy can be readily overcome through the exchange of the antibiotic partner. With the anticipated spread of plasmid-mediated colistin resistance, we propose further investigation into colistin combination therapy as a potential last resort therapeutic option.

## Results

**Colistin potentiates antibiotics in *mcr-1*-positive bacteria.** To date, all clinical and environmental isolates expressing *mcr-1* have been members of the Enterobacteriaceae and, as such, many of these pathogens, in addition to *Pseudomonas aeruginosa*, have been shown to capably express *mcr-1* in a laboratory setting[7]. *E. coli*, *Salmonella* Typhimurium, *Enterobacter aerogenes*, *Enterobacter cloacae*, and *K. pneumoniae* are all major pathogens of clinical importance known to harbor *mcr-1*. Therefore, these species were engineered to express *mcr-1* from the pGDP2 plasmid, and investigated for susceptibility to colistin combination treatment. We screened > 40 antibiotics covering all major drug classes for changes in MIC in the presence of colistin. Fold reduction in MIC was quantified by dividing the MIC of an antibiotic alone by its MIC in the presence of a therapeutic concentration of colistin (2 μg mL$^{-1}$). As expected, *mcr-1* confers no change in susceptibility (<4-fold change) to antibiotics outside of the polymyxin class (Supplementary Table 1). However, in the presence of colistin, several antibiotics are highly potentiated, as characterized by a greater than eightfold reduction in MIC (Fig. 1, Supplementary Table 1). Importantly, the 2 μg mL$^{-1}$ colistin concentration used when determining fold reduction in MIC represents a concentration obtainable during standard therapeutic colistin dosing[27–29]. In combination with colistin, the antibiotics rifampicin, rifabutin, clarithromycin, minocycline, and novobiocin achieve the highest therapeutic potential as combinatorial partners, with an observed reduction in MIC below the corresponding Gram-positive clinical breakpoint for all Enterobacteriaceae tested. Of note, with the removal of novobiocin from market in 2011, there is no currently listed CLSI or EUCAST breakpoint. However, we classify novobiocin as having high clinical potential as a combination agent with colistin, as it is

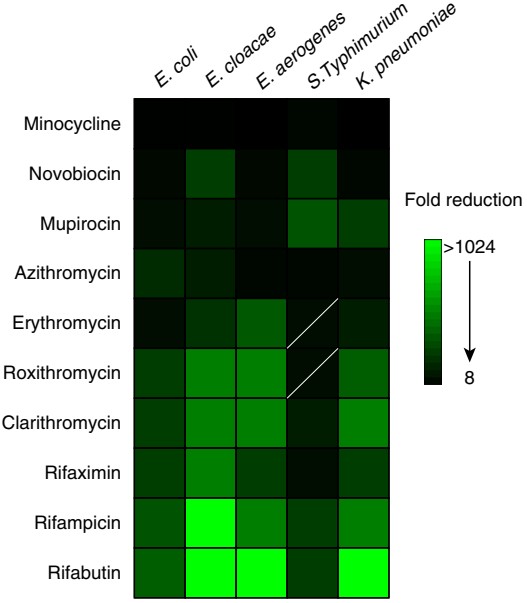

**Fig. 1** Colistin potentiates antibiotics conventionally used against Gram-positive bacteria in Enterobacteriaceae expressing *mcr-1*. Heat map showing the mean fold reduction of MIC in the presence of 2 µg mL$^{-1}$ colistin for strains transformed with pGDP2:*mcr-1*. Antibiotics listed were potentiated ≥ 8-fold across all lab generated Enterobacteriaceae strains. A lack of potentiation below clinical breakpoint is indicated by a diagonal white line. Data are representative of two biological replicates

potentiated to active concentrations below steady-state serum levels (5 µg mL$^{-1}$)[30]. Indeed, the unique mechanism of action and potential of novobiocin as a combination partner with colistin and other Gram-negative outer membrane disrupting compounds[25] may justify a re-evaluation of clinical use. Additionally, the topical antibiotic mupirocin is potentiated to clinically pertinent levels of susceptibility, representing a potential treatment strategy for the increasing threat of multidrug-resistant skin and soft-tissue infections[31].

To investigate whether colistin potentiation is conserved beyond laboratory-generated *mcr-1* strains, we tested nine clinical and retail food-derived *mcr-1*-positive *E. coli* strains with clarithromycin, novobiocin, and rifampicin. Encouragingly, all nine strains were susceptible to potentiation, with an observed fold reduction in MIC greater than eight for all three antibiotics (Supplementary Fig. 1). Susceptibility below clinical breakpoint is achieved in the presence of colistin for clarithromycin or rifampicin in six of nine and, for novobiocin, in seven of nine strains. Overall, eight of nine strains demonstrate therapeutic levels of sensitivity to at least one of the three antibiotic-colistin combinations. In addition to the broad conservation of susceptibility to colistin combination treatment observed across the *mcr-1*-positive Enterobacteriaceae species tested, potentiation was highly dependent upon antibiotic class. Notably, broad-spectrum antibiotics, such as those within the fluoroquinolone and beta-lactam classes, displayed a limited reduction in MIC and no impact on clinical breakpoints. Antibiotics in the rifamycin and macrolide class, as well as minocycline and novobiocin, displayed the highest levels of synergy with colistin in *mcr-1*-positive Enterobacteriaceae.

To determine if expression of *mcr-1* increases resistance to the outer-membrane disruption activity of colistin proportionally to the increased resistance to growth-inhibition, we investigated the concentration range of colistin capable of potentiating rifampicin (1 µg mL$^{-1}$) into wild-type and *mcr-1*-expressing *E. coli*. Due to the physicochemical properties of rifampicin, outer-membrane

disruption is essential to facilitate entry and growth-inhibition. In a wild-type background, growth-inhibition occurs at a colistin concentration of 0.15 µg mL$^{-1}$ and decreases twofold to 0.075 µg mL$^{-1}$ in the presence of rifampicin at 1 µg mL$^{-1}$ (Fig. 2a). Therefore, while a concentration of 0.075 µg mL$^{-1}$ colistin is insufficient to inhibit bacterial growth, it provides sufficient outer-membrane disruption to allow rifampicin uptake and rifampicin-mediated growth-inhibition. Expression of *mcr-1* increases the concentration of colistin required for growth-inhibition to 5 µg mL$^{-1}$ (Fig. 2b), which is 32-fold greater than wild-type. However, in the presence of rifampicin, the concentration of colistin required to inhibit growth is 0.15 µg mL$^{-1}$ (Fig. 2b) or only twofold greater than the concentration required to potentiate rifampicin in wild-type cells. Thus, expression of *mcr-1* appears to provide significant protection against the monotherapy activity of colistin, but not its capacity to perturb the outer membrane.

To further characterize the impact of *mcr-1* on the outer-membrane perturbation activity of colistin, we measured uptake of the hydrophobic fluorophore N-phenyl-1-naphthylamine (NPN). An intact outer membrane prevents entry of NPN to the phospholipid layer and the subsequent fluorescence. Therefore, NPN uptake represents a quantitative read-out for colistin-mediated outer-membrane disruption. We observed only a 2- to 4-fold increase in colistin concentrations required to reach comparable levels of NPN uptake in *mcr-1*-expressing *E. coli* compared to wild-type (Fig. 2c). For example, initial uptake (>5%) occurs at 0.39 µg mL$^{-1}$ of colistin in wild-type *E. coli* and 0.78 µg mL$^{-1}$ in an *mcr-1* background (Fig. 2c). Saturation of NPN uptake (>95%) occurs at 6.25 µg mL$^{-1}$ in wild-type and 12.5 µg mL$^{-1}$ when expressing *mcr-1* (Fig. 2c). All additional levels of NPN uptake observed in a wild-type background between the initial uptake and saturation can be achieved by increasing the concentration of colistin 2- to 4-fold in an *mcr-1*-expressing strain (Fig. 2c). Despite a 32-fold change in MIC observed with expression of *mcr-1*, the concentration of colistin required for significant outer-membrane perturbation is only increased 2- to 4-fold.

With the observed discrepancy between susceptibility to outer-membrane perturbation and resistance to growth-inhibition conferred by *mcr-1*, we hypothesized *mcr-1* may provide resistance through a mechanism beyond reducing the initial electrostatic interaction with the outer membrane. To determine if *mcr-1* expression alters the rate at which colistin-mediated lysis occurs, we monitored the reduction of optical density (OD) over 18 h in wild-type and *mcr-1*-expressing *E. coli*. To isolate the impact of *mcr-1* on lytic rate and normalize for differences in strain susceptibility, we compared the lowest concentration of colistin capable of causing growth-inhibition after 18 h when the initial inoculum is ~ 8 × 10$^8$ colony-forming units (CFU) mL$^{-1}$. Typically, exposure to inhibitory concentrations of colistin results in rapid bacterial lysis and, therefore, a reduction in culture turbidity. Indeed, wild-type *E. coli* displayed an exponential decrease from the starting ~ 0.5 OD (600 nm) over 6 h with clearance plateauing at ~ 0.1 OD (600 nm) (Fig. 2d). However, expression of *mcr-1* drastically reduced the rate of bacterial lysis, with OD (600 nm) increasing over the first 3 h followed by a slow decline to ~ 0.4 OD (600 nm) after 18 h (Fig. 2d). Notably, we were unable to return the rate of lysis to wild-type levels by increasing the concentration of colistin (Supplementary Fig. 2a, b). The observed change in lytic susceptibility appears to be polymyxin specific, as lysis by ampicillin occurs as previously described[32] for both strains (Supplementary Fig. 2c).

**Overcoming resistance to colistin combination therapy.** In cases where colistin combination treatment effectively inhibits the

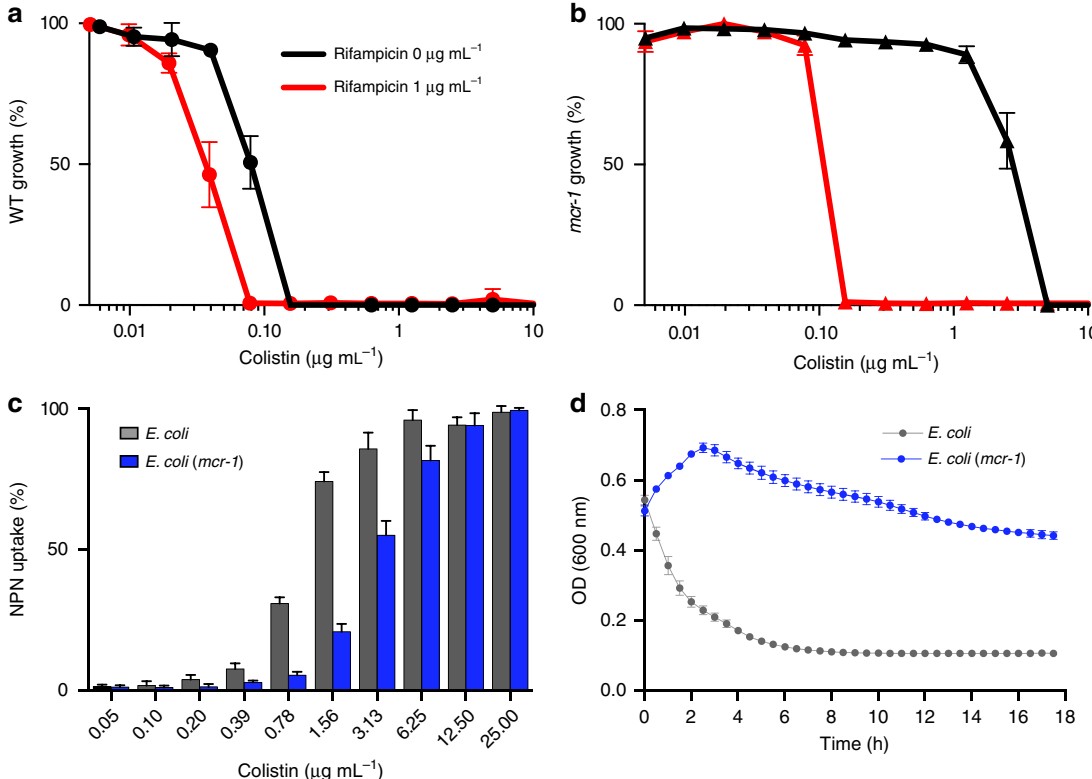

**Fig. 2** Expression of *mcr-1* provides limited protection to colistin-mediated outer-membrane disruption. **a**, **b** Potency analysis of wild-type (**a**) and *mcr-1*-expressing (**b**) *E. coli* in the presence (red) and absence (black) of rifampicin (1 μg mL$^{-1}$). **c** *N*-Phenyl-1-naphthylamine (NPN) uptake of wild-type (gray) and *mcr-1*-expressing *E. coli* (blue) induced by colistin. NPN uptake (%) represents the background subtracted fluorescence divided by the fluorescence observed at 100 μg mL$^{-1}$ of colistin. **d** Kinetic analysis of colistin-mediated lysis in wild-type (gray) and *mcr-1*-expressing (blue) *E. coli*. Optical density (OD) at 600 nm was monitored every 30 min for 18 h in the presence of colistin at 50 μg mL$^{-1}$ and 6.25 μg mL$^{-1}$ in *mcr-1* and wild-type, respectively. Concentrations selected are the lowest values capable of inhibiting growth with a starting cell density of OD (600 nm) 0.5. Data in **a**, **b**, **c**, and **d** represent means with standard deviation from two biological replicates

growth of an *mcr-1*-positive strain, antimicrobial activity is driven solely by the antibiotic partner. Consequently, reduced efficacy in this form of treatment would likely be facilitated by resistance to the antibiotic partner, rather than to the potentiation ability of colistin[33]. As such, colistin combination therapy offers the unique advantage that, despite the potential for the evolution of resistance, substitution of the antibiotic partner should renew treatment efficacy. Indeed, spontaneous *E. coli* mutants expressing *mcr-1* generated in the presence of rifampicin and colistin were no longer susceptible to colistin/rifampicin combination treatment but remained susceptible if rifampicin was exchanged for either novobiocin or clarithromycin at therapeutically relevant levels (Fig. 3a, b and Supplementary Table 2).

**In vivo efficacy**. Given the immediate therapeutic potential of combination therapy, we investigated clarithromycin in combination with colistin using two murine models of *mcr-1*-positive *K. pneumoniae* infection. Recognizing the potential for dose-sparing during drug combination therapies and noted concerns of colistin toxicity, we tested the efficacy of colistin at approximately one fifth of the human equivalent dose[34]. Clarithromycin was selected for its high propensity for in vitro potentiation among all Enterobacteriaceae tested (Fig. 1) and dosed at approximately the standard human equivalent dose[35]. During a thigh infection, neither colistin (7.5 mg kg$^{-1}$) nor clarithromycin (200 mg kg$^{-1}$) showed a significant reduction in bacterial load when treatment was initiated 1 h post infection (Fig. 4a). However, the combination of colistin and clarithromycin proved efficacious, resulting in a 2.9-log$_{10}$ reduction ($p < 0.0001$, Mann–Whitney *U*-test) in

CFU when compared to the untreated control 8 h after infection (Fig. 4a). *K. pneumoniae* bloodstream infections cause a high level of patient mortality and, therefore, represent a prime candidate for novel last resort therapeutics. To produce a murine bacteremia infection, mice were inoculated with a dose of *mcr-1*-expressing *K. pneumoniae* that led to 100% lethality within 12 h. Monotherapy treatments administered 1 h post infection did not demonstrate significant survival beyond that of the untreated group (Fig. 4b). Importantly, within 1 h post infection, *mcr-1*-expressing *K. pneumoniae* was detected at high burdens throughout the animal (Supplementary Fig. 3). Animals receiving clarithromycin and colistin combination therapy daily for 5 days rescued 60% of those treated (Fig. 4b), highlighting the in vivo efficacy of colistin combination therapy against *mcr-1*-positive infections.

## Discussion

A diminishing antibiotic pipeline and increasing clinical reliance on colistin has magnified the threat of horizontally transferrable colistin resistance via the *mcr-1* gene. Similar to many other cationic peptides, the impact of colistin on bacterial cells can be broadly classified into three major events, outer-membrane disruption through interaction with surface LPS, self-promoted uptake through the outer membrane, and the formation of regions of instability in the cytoplasmic membrane leading to lysis[1,2,36]. Previous studies have leveraged the outer-membrane disruptive activity of colistin by combining it with antibiotics that otherwise are unable to traverse an intact outer membrane[10–16]. Notably, this susceptibility to combination therapy is maintained

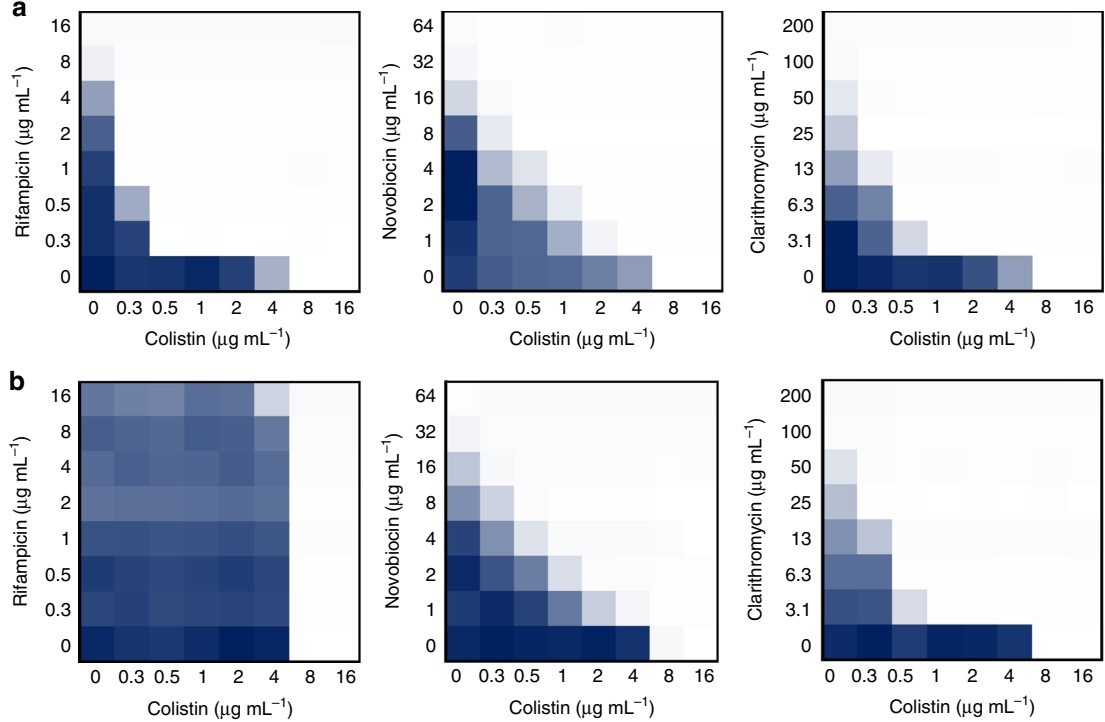

**Fig. 3** Resistance to colistin combination therapy can be overcome by exchange of the partnered antibiotic. **a,b** Checkerboard broth microdilution assays showing dose-dependent potentiation of rifampicin, novobiocin, and clarithromycin by colistin against *mcr-1*-positive *E. coli* (**a**) and a spontaneous mutant of *E. coli*-expressing *mcr-1* (**b**) generated in the presence of rifampicin and colistin. Dark blue regions represent higher cell density. Data in **a** and **b** represent the mean OD (600 nm) of two biological replicates

in strains with a number of conventional chromosomal colistin resistance mechanisms[19–21].

As *mcr-1* represents the predicted driver of pervasive colistin resistance, we investigated the potential of combination therapy for the treatment of pathogens expressing *mcr-1*. Our data shows that *mcr-1*-expressing Enterobacteriaceae can be sensitized to a range of antibiotics in the presence of colistin. Importantly, sensitization occurs at concentrations below clinical breakpoint for colistin as well as the partnered antibiotic and is efficacious in treating murine models of *mcr-1*-expressing *K. pneumoniae*. The amenability of this form of combination therapy to the unconventional use of narrow-spectrum, Gram-positive active antibiotics poses an advantage over traditional broad-spectrum approaches. Specifically, Gram-negative pathogens are unlikely to be harboring the appropriate intrinsic resistance mechanisms that would render combination treatment with such Gram-positive active antibiotics ineffective, due to a lack of selective pressure. However, should resistance develop to combination treatment, our results suggest that resensitization may be achieved with an exchange of the antibiotic partnered with colistin.

The traditional focus of the proposed mechanisms for *mcr-1* and other polymyxin resistance is that the addition of cationic groups to the phosphates of lipid A reduces the electrostatic interaction between colistin and lipid A, preventing localized disruption of the outer membrane and, therefore, self-promoted uptake and lysis[3,7,37,38]. However, this proposed mechanism would not appear to predict the observed susceptibility of *mcr-1*-expressing bacteria to outer-membrane disruption. It is interesting to note the pEtN modification of *mcr-1* provides only limited reduction to the formal charge of the phosphate ester of lipid A from −1.5 to −1[39]. Despite the significant increase in the concentration of colistin required for growth-inhibition in an *mcr-1*-expressing strain, both our rifampicin susceptibility and NPN uptake data show that colistin is able to interact with and disrupt

the outer membrane sufficiently to allow the uptake of large hydrophobic compounds largely irrespective of *mcr-1* expression. Additionally, we demonstrate that *mcr-1* not only increases the required colistin concentration to inhibit growth, but also reduces the rate at which lysis occurs. With continued susceptibility to outer-membrane perturbation and reduction in the rate of colistin-mediated bacterial lysis, we hypothesize that despite the monogenetic nature of *mcr-1*, colistin resistance is conferred through a mechanism that extends beyond decreasing the strength of the electrostatic interaction between colistin and lipid A in the outer membrane.

The relationship between bactericidal activity and the ability of polymyxins to interact with and disrupt the outer membrane is poorly characterized. Indeed, a growing range of evidence, including the data presented in this study, suggests these two processes may not be highly linked[40,41]. For example, in wild-type backgrounds, outer-membrane disruption can be achieved with PMBN[42], which lacks the fatty-acyl moiety found on polymyxin B and colistin, and is incapable of inducing lysis. The lytic antimicrobial activity of colistin is thought to require insertion of the fatty-acyl chain into the outer membrane[42], which weakens the packing of adjacent lipid A molecules and facilitates transit to the inner membrane via self-promoted uptake. While the mechanism behind self-promoted colistin uptake is not yet fully elucidated, this step is crucial in advancing from outer-membrane disruption to lysis[43]. One potentially underappreciated aspect of *mcr-1* that may contribute to its ability to resist the growth-inhibitory activity of colistin is that the addition of cationic groups like pEtN can alter outer-membrane architecture through a reduction in repulsion between neighboring LPS molecules that strengthens membrane packing[39,44]. It has been speculated the high intrinsic polymyxin resistance observed in *N. meningitides* can be attributed to this phenomenon[39]. Notably, the main component of polymyxin resistance in *N. meningitides* is the

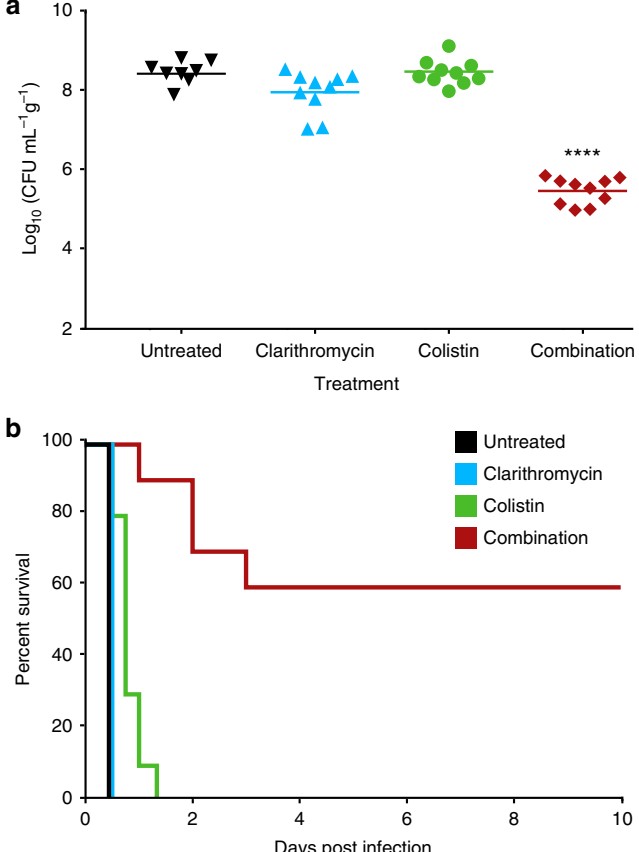

**Fig. 4** Colistin and clarithromycin combination therapy is efficacious in two mouse models of infection. **a** Single-dose treatment at 1 h post infection of clarithromycin ($n = 10$, blue, 200 mg kg$^{-1}$, p.o.), colistin ($n = 10$, green, 7.5 mg kg$^{-1}$, i.p.) or the combination ($n = 10$, red) in a neutropenic mouse thigh infection model using *mcr-1*-expressing *K. pneumoniae*. Colony-forming units (CFU) within thigh tissue were enumerated at 8 h post infection and compared to the untreated group ($n = 8$, black). Horizontal lines represent geometric mean of the bacterial load for each treatment group. The combination of colistin and clarithromycin resulted in a 2.9-log$_{10}$ reduction ($p < 0.0001$, Mann–Whitney *U*-test) in CFU when compared to the untreated control 8 h after infection. **b** Survival curve of a *K. pneumoniae*-expressing *mcr-1* bacteremia infection dosed at 1, 24, 48, 72, 96, and 120 h post infection as outlined above for clarithromycin ($n = 10$), colistin ($n = 10$), untreated ($n = 10$), and the combination ($n = 10$)

phosphoethanolamine transferase LptA[45], which is closely related to the structure of MCR-1[7]. Therefore, we hypothesize that strengthened LPS packing provided by *mcr-1* could play an important role in reducing the uptake and lytic activities of colistin.

The ability of colistin to potentiate antibiotics against colistin-resistant bacteria is not well understood. Interestingly, expression of *mcr-1* reduces the potentiation ability of PMBN[25] but does not alter appreciably the susceptibility to the outer-membrane disruption of colistin. Thus, it may be that the role of the fatty-acyl tail in self-promoted uptake and lysis is impaired by *mcr-1* but its importance in outer-membrane disruption is unaffected. Interestingly, many of the toxicity concerns associated with polymyxins can be attributed to this fatty-acyl tail[42] and changes or elimination of this hydrophobic moiety may represent a challenging avenue to combination therapy strategies in the face of *mcr-1*-mediated resistance. Overall, exploiting the susceptibility of *mcr-1*-expressing pathogens to colistin combination therapy may represent an Achilles heel for an otherwise difficult resistance

mechanism, and we recommend a re-evaluation of its clinical utility.

## Methods

**Colistin combination susceptibility testing**. Fourteen Enterobacteriaceae strains harboring *mcr-1* were investigated for susceptibility to colistin combination therapy. Specifically, *E. coli* K-12 BW25113[46], *K. pneumoniae* ATCC 43816, *S.* Typhimurium ATCC 14028, *E. cloacae* GN687 (Public Health Ontario), and *E. aerogenes* C0064 (Public Health Ontario) were transformed with the pGDP2:*mcr-1* plasmid. Additionally, the National Microbiology Laboratory of Canada provided *mcr-1*-positive clinical and retail food-derived *E. coli* isolates; N12-00130 (ST624), N15-02865 (ST648), N15-02866 (ST398), N16-00121 (ST3944), N16-00319 (ST156), N16-00487 (ST648), N16-01175 (ST515), N16-03711 (ST10), and N17-00323 (ST101). Lab generated strains were validated for colistin resistance above EUCAST breakpoint using Clinical and Laboratory Standard Institute (CLSI) guidelines for MIC[47]. Fold reduction of MIC was determined by dividing the MIC of the antibiotic alone by its MIC in the presence of 2 µg mL$^{-1}$ colistin. Compounds demonstrating ≥ 8-fold reduction were prioritized based on sensitization below their corresponding Enterobacteriaceae or *Staphylococcus aureus* clinical breakpoint. Checkerboard analyses were conducted with each drug serially diluted at eight concentrations to create an 8 × 8 matrix. At least two biological replicates were done for each combination and the means were used for FIC calculation. The FIC for each drug was calculated by dividing the concentration of drug in the presence of codrug in a combination for a well showing < 10% growth, by the MIC for that drug alone[48]. The FIC index is the sum of the two FICs, with an FIC index ≤ 0.5 deemed synergistic.

**Membrane integrity assays**. All assays were performed with wild-type and pGDP2:*mcr-1*-expressing *E. coli* BW25113. MICs were conducted as outlined above in the presence and absence of 1 µg mL$^{-1}$ rifampicin. NPN assays were conducted as previously established[49]. Cells from an overnight culture were diluted 1/50 and incubated until mid-log (~ 0.5 OD 600 nm), centrifuged, washed in 5 mM HEPES buffer containing 20 mM glucose, spun down and resuspended in the same buffer to an OD (600 nm) of 1. A volume of 100 µL of cells was added to 100 µL of buffer containing NPN and varying concentrations of colistin in black clear-bottom 96-well plates. After a 1 h incubation at room temperature fluorescence was read in a Tecan® infinite M1000 Pro, excitation 355 ± 5 nm and emission 420 ± 5 nm. Percent NPN uptake is calculated for each strain according to ref. [50]

$$\text{NPN uptake}\,(\%) = (F_{\text{obs}} - F_0)/(F_{100} - F_0) \times 100\%$$

where $F_{\text{obs}}$ is the observed fluorescence at a given colistin concentration, $F_0$ is the initial fluorescence of NPN with *E. coli* cells in the absence of colistin, and $F_{100}$ is the fluorescence of NPN with *E. coli* cells upon addition of 100 µg mL$^{-1}$ of colistin, which is beyond the observed plateau in fluorescence for both strains.

To determine the rate of colistin-induced lysis, strains were grown to mid-log (2–3 h), centrifuged, suspended in fresh Mueller Hinton broth to an OD (600 nm) of 1 and added to varying concentrations of colistin to a total volume of 250 µL in a standard 96-well plate, giving a starting OD (600 nm) of ~ 0.5 in the plate reader. OD (600 nm) was measured every 30 min for 18 h during incubation at 37 °C with shaking using a Tecan® Sunrise.

**Generation of combination-resistant mutants**. *E. coli* BW25113 transformed with pGDP2:*mcr-1* was used to generate mutants resistant to rifampicin and colistin combination therapy through the plating of bacteria onto 1 µg mL$^{-1}$ colistin and 2 µg mL$^{-1}$ rifampicin, and incubated until colony growth. Single colonies were passaged overnight in rifampicin and colistin to reconfirm resistance prior to investigation with MIC, and fold change assays, which were performed as outlined above.

**Animal studies**. All animal studies were conducted according to guidelines set by the Canadian Council on Animal Care using protocols approved by the Animal Review Ethics Board at McMaster University under Animal Use Protocol # 17-03-10. All animal studies were performed with 6–8-week-old female CD-1 mice from Charles River. Female mice were used in accordance with previously established models[51] as well as ease of housing and randomization. Sample size was selected based on the results of a preliminary infection trial. Before infection, mice were relocated at random from a housing cage to treatment or control cages. No animals were excluded from analyses and blinding was considered unnecessary.

**Mouse thigh infection model**. The combination of colistin and clarithromycin was tested against *K. pneumoniae* ATCC 43816 transformed with pGDP2:*mcr-1* in a neutropenic mouse thigh infection model. Female CD-1 mice were rendered neutropenic by cyclophosphamide, dosed at 150 and 100 mg kg$^{-1}$ delivered on days -4 and -1 prior to infection. Bacteria were suspended in sterile saline and adjusted to a concentration of ~ 1 × 10$^6$ CFU per infection site and injected into the right and left thighs of five mice per treatment group. At 1 h post infection, mice received either colistin (7.5 mg kg$^{-1}$, i.p. $n = 10$), clarithromycin (200 mg kg$^{-1}$, p.o.

$n = 10$), untreated ($n = 8$), or the combination ($n = 10$). Mice were euthanized 8 h after infection, thigh tissue was aseptically collected, weighed, homogenized, serially diluted in PBS and plated onto solid LB supplemented with kanamycin ($50 \mu g \, mL^{-1}$). Plates were incubated overnight at 37 °C and colonies were quantified to determine bacterial load.

**Mouse bacteremia model.** The combination of colistin and clarithromycin was tested against *K. pneumoniae* ATCC 43816 transformed with pGDP2:*mcr-1* in an immunocompetent bacteremia infection model. Female CD-1 mice were infected intraperitoneally with ~ $1 \times 10^6$ CFU of bacteria with 5% porcine mucin (Sigma-Aldrich). Infections were allowed to establish for 1 h prior to treatment with colistin, clarithromycin, or the combination. With the encouraging reduction in CFU observed in the thigh infection model, dosing was administered as described above. Clinical endpoint was determined using a five-point body condition score analyzing weight loss, decrease in body temperature, respiratory distress, hampered mobility, and hunched posture. Experimental endpoint was defined as 10 days post infection for mice not reaching clinical endpoint.

**Data availability**. The data that support the findings of this study are available in this article and its Supplementary Information files, or from the corresponding author upon request.

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

## Acknowledgements

We would like to thank Laura Mataseje for assistance at the NML and Drs. Reid-Smith, Rubin, Toye, and Zhanel for supplying *mcr-1* isolates. We would also like to thank Dr. Roberto Melano at Public Health Ontario for bacterial strains GB687 and C0064. This research was supported by a Foundation grant from the Canadian Institutes for Health Research (FDN-143215) to E.D.B. as well as a grant from the Ontario Research Fund (Research Excellence program), and by a salary award to E.D.B. from the Canada Research Chairs Program. C.R.M. was supported by a Canadian Institutes for Health Research scholarship. L.A.C. was supported by a Canadian Institutes for Health Research scholarship. A.A.F.-C. was supported by the Ontario Research Foundation grant.

## Author contributions

C.R.M., J.M.S., L.A.C., M.R.M., and E.D.B. designed the experiments. C.R.M. and J.M.S. designed and conducted the in vitro potentiation assays. M.R.M. provided access to clinical and food-derived *mcr-1*-expressing *E. coli* isolates and assisted in colistin potentiation screening. C.R.M. and L.A.C. designed and performed in vivo infection model experiments. B.K.C. and A.A.F.-C. supported murine model development. C.R.M. and E.D.B. wrote the manuscript with input from all authors.

## Additional information

**Competing interests:** The authors declare no competing financial interests.

