## [Peer Review File · Nature Communications]

Reviewers' comments:

Reviewer #1 (Remarks to the Author):

This is a well-constructed piece of work, and the methodology is precise, but it is unclear if the message has sufficient novelty to be presented in Nature Communications as an original information of potential broad interest. Certainly, this manuscript could be well received in more specific journals devoted to antimicrobial agents, anti-infective therapy or even infectious diseases.

The current title is attractive, but to a certain extent unprecise and misleading. Titles as: "...using antibiotic combinations with colistin", or "...adding antibiotics to colistin" are less sexy but more accurate.

The fact that polymyxins acts as permeating agents allowing the internalization of other antibiotic drugs is a very well-known fact in the literature. The permeabilizing effect of colistin making possible "synergistic associations" with rifampicin and macrolides is rightly quoted by the authors.

The main contribution of this manuscript is that the mechanism of resistance mediated by the gene *mcr-1* do not eliminate this permeabilizing activity. That is not entirely surprising, as also chromosomal colistin resistance do not remove colistin potentiation of the activity of other drugs (that is also rightly mentioned by the authors).

Note that probably colistin might influence the surface even of intrinsically resistant organisms, as *Staphylococcus aureus* (Effects of colistin on biofilm matrices of *Escherichia coli* and *Staphylococcus aureus*. Klinger-Strobel M, et al. *Int J Antimicrob Agents*. 2017 49:472-479), some recent research in progress (in press?) demonstrate the effect of colistin in overcoming even Gram-positive and *Mycobacterium* intrinsic resistance.

The antibiotics that the author's show to permeate (and thus being active) in the presence of colistin obviously include those (generally with high molecular weight) that are excluded by the outer-membrane of Gram negatives, as clarithromycin or novobiocin.

The use of several Enterobacteria to test the universality of the principle is OK, but not really essential. Moreover, particularly for the *mcr-1* positive clinical and retail food derived *E. coli* isolates, some information about clonality will be welcome to assure that the same clone has not been tested twice.

The sentence line 126 should be corrected: "antibiotics mupirocin and fusidic acid are also highly potentiated with mupirocin...". Mupirocin potentiates mupirocin?

Line 142. The fact that polymyxins permeate minocycline is an "historical" finding (Chopra, I., & Hacker, K. (1992). Uptake of minocycline by *Escherichia coli*. *Journal of Antimicrobial Chemotherapy*, 29(1), 19-25).

Reviewer #2 (Remarks to the Author):

This is an interesting paper exploring synergy of colistin, a cationic lipopeptide antibiotic with activities against Gram-negative bacteria, with a large panel of antibiotics including Gram-positive agents, against *E. coli* and several other Enterobacteriaceae species strains producing MCR-1, a plasmid-mediated phosphoethanolamine transferase which confers resistance to colistin. The authors show that in combination with colistin at 2 mg/L, rifampicin, rifabutin, clarithromycin, minocycline and novobiocin achieve the highest therapeutic potential as combinatorial partners, with the MICs of these agents mostly reaching below the susceptibility breakpoints of these agents

for Gram-positive pathogens. The work is nicely executed, and is further supported by limited efficacy data obtained by two mouse infection models. Two key limitations identified are that this phenomenon (potentiation of Gram-positive antibiotics by colistin against Gram-negative bacteria) has been well documented if not for *mcr-1*, and that mechanisms underlying the observed synergy are speculated but not explored experimentally.

Comments:

-While the use of 2 mg/L of colistin seems reasonable, it should be noted that significant individual variations exist in the steady-state colistin concentrations in the plasma, and whether this concentration is "readily obtainable during standard therapeutic colistin dosing" is a subject of debate.

-How was dosing in the thigh infection model determined?

-“The susceptibility to potentiation associated with *mcr-1* mediated colistin resistance may represent an Achilles heel for an otherwise devastating resistance mechanism” – this case is cannot be made unless lack of such synergy is not observed in *mcr-1*-negative strains. This reviewer’s understanding that the synergy can be observed in either scenario.

L203: Please include a reference for the FIC index.

L216: Why were only female mice used?

L220: Please include references for the antibiotic doses used.

L225: Technically, this appears to be an intra-abdominal infection model (though there is indeed secondary bacteremia).

L335: What is the rationale for Supplemental Figure 2? The point seems to be the sterility of spleen post-treatment, but there is no parallel 10-day control shown (only 1-hour control is shown).

Reviewers' comments:

Reviewer #1 (Remarks to the Author):

This is a well-constructed piece of work, and the methodology is precise, but it is **unclear if the message has sufficient novelty to be presented in Nature Communications** as an original information of potential broad interest. Certainly, this manuscript could be well received in more specific journals devoted to antimicrobial agents, anti-infective therapy or even infectious diseases.

We appreciate the reviewer's concerns. To address novelty, we have expanded on the mechanistic details behind the susceptibility of *mcr-1* to colistin combination therapy. Additionally, we explore the discrepancy between susceptibility to colistin mediated outer-membrane perturbation and resistance to growth-inhibition in *mcr-1* expressing strains. Specifically, these aspects are addressed in Figure 2a-d, Lines 175 to 227 and Lines 292 to 344. We feel this information contributes to furthering our understanding of *mcr-1* resistance as well as provides insight into the ability of colistin to potentiate antibiotics against colistin resistant bacteria. Notably, this information can assist in furthering our understanding of polymyxin resistance and will help guide future therapeutic discovery efforts. As such this work is immediately relevant to the wide readership of *Nature Communications*, including antibiotic drug discoverers, clinicians, and a broad community of engaged individuals who are increasingly calling for solutions to the antimicrobial resistance crisis.

The current title is attractive, but to a certain extent unprecise and misleading. Titles as: ".....using antibiotic combinations with colistin", or "....adding antibiotics to colistin" are less sexy but more accurate.

We thank the reviewer for their thoughts regarding the title and sympathize with these sentiments though we don't have a particularly strong opinion on this. We recognize that this question is reasonably in the domain of the journal's editorial staff and have not revised the title at this time. We will defer to their best judgement should the work be accepted for publication.

The fact that polymyxins acts as permeating agents allowing the internalization of other antibiotic drugs is a very well-known fact in the literature. The permeabilizing effect of colistin making possible "synergistic associations" with rifampicin and macrolides is rightly quoted by the authors.

We thank the reviewer for this comment.

The main contribution of this manuscript is that the mechanism of resistance mediated by the gene *mcr-1* do not eliminate this permeabilizing activity. That is not entirely surprising, as also chromosomal colistin resistance do not remove colistin potentiation of the activity of other drugs (that is also rightly mentioned by the authors).

We thank the reviewer for this comment and believe we have further expanded the contributions of this manuscript as outlined above. Additionally, we have outlined differences between *mcr-1* mediated colistin resistance and traditional chromosomal resistance mechanisms in Lines 89 to 109.

Note that probably colistin might influence the surface even of intrinsically resistant organisms, as *Staphylococcus aureus* (Effects of colistin on biofilm matrices of *Escherichia coli* and *Staphylococcus aureus*. Klinger-Strobel M, et al. Int J Antimicrob Agents. 2017 49:472-479), some recent research in

progress (in press?) demonstrate the effect of colistin in overcoming even Gram-positive and Mycobacterium intrinsic resistance.

We appreciate this comment and have included a statement on intrinsically resistant organism susceptibility to colistin Lines 85 to 87 and included the recommended reference.

The antibiotics that the author's show to permeate (and thus being active) in the presence of colistin obviously include those (generally with high molecular weight) that are excluded by the outer-membrane of Gram negatives, as clarithromycin or novobiocin.

The observation made by the reviewer is appreciated and has been included in Lines 113 to 115 and Lines 302 to 303.

The use of several Enterobacteria to test the universality of the principle is OK, but not really essential. Moreover, particularly for the mcr-1 positive clinical and retail food derived E. coli isolates, **some information about clonality will be welcome to assure that the same clone has not been tested twice.**

We appreciate the concern regarding clonality and have included ST types for the strains tested in Lines 353 to 355. Strains N15-02865 and N16-00487 are both ST648 however core genome analysis revealed 1647 SNVs.

The sentence line 126 should be corrected: "antibiotics mupirocin and fusidic acid are also highly potentiated with mupirocin...". Mupirocin potentiates mupirocin?

We thank the reviewer for identifying this error and have corrected the sentence in Lines 154 to 156.

Line 142. The fact that polymyxins permeate minocycline is an "historical" finding (Chopra, I., & Hacker, K. (1992). Uptake of minocycline by Escherichia coli. Journal of Antimicrobial Chemotherapy, 29(1), 19-25).

We appreciate the reviewer comment and have added minocycline to Line 84 and added the recommended reference.

Reviewer #2 (Remarks to the Author):

This is an interesting paper exploring synergy of colistin, a cationic lipopeptide antibiotic with activities against Gram-negative bacteria, with a large panel of antibiotics including Gram-positive agents, against E. coli and several other Enterobacteriaceae species strains producing MCR-1, a plasmid-mediated phosphoethanolamine transferase which confers resistance to colistin. The authors show that in combination with colistin at 2 mg/L, rifampicin, rifabutin, clarithromycin, minocycline and novobiocin achieve the highest therapeutic potential as combinatorial partners, with the MICs of these agents mostly reaching below the susceptibility breakpoints of these agents for Gram-positive pathogens. The work is nicely executed, and is further supported by limited efficacy data obtained by two mouse infection models. **Two key limitations identified are that this phenomenon (potentiation of Gram-positive antibiotics by colistin against Gram-negative bacteria) has been well documented if not for mcr-1, and that mechanisms underlying the observed synergy are speculated but not explored experimentally.**

We appreciate the reviewer's concerns. 1) In the revised manuscript, we have endeavored to better articulate the important differences between *mcr-1* mediated colistin resistance and traditional chromosomal resistance in Lines 89 to 109. Additionally, we show differences in the potentiation activity of colistin in wild type and *mcr-1* expressing *E. coli* (Fig. 2a,b). 2) As outlined above, we have expanded on the mechanistic details behind the susceptibility of *mcr-1* to colistin combination therapy through the addition of multiple experiments.

Comments:

-While the use of 2 mg/L of colistin seems reasonable, it should be noted that significant individual variations exist in the steady-state colistin concentrations in the plasma, and whether this concentration is “**readily obtainable during standard therapeutic colistin dosing**” is a subject of debate.

We appreciate the reviewer's comments and concern regarding plasma availability and we have adjusted Lines 142 to 144 and added additional references.

-How was dosing in the thigh infection model determined?

We thank the reviewer for their interest in dose selection and have included this information in Lines 245 to 249.

-“The susceptibility to potentiation associated with *mcr-1* mediated colistin resistance may represent an Achilles heel for an otherwise devastating resistance mechanism” – this case is cannot be made unless lack of such synergy is not observed in *mcr-1*-negative strains. This reviewer's understanding that the synergy can be observed in either scenario.

We appreciate the reviewers comment on Lines 341 to 344. However, we feel with the additional data included in the manuscript (Fig 2a,b,c) this statement remains appropriate. The inability of *mcr-1* to provide resistance to colistin mediated outer membrane disruption is an exploitable weakness and shortcoming of this resistance mechanism that can be leveraged into a potential treatment option.

L203: Please include a reference for the FIC index.

A reference has been included for the FIC calculation in Line 366.

L216: Why were only female mice used?

We appreciate the reviewers comment on sex of the animals used and we have addressed this in Lines 404 to 406.

L220: Please include references for the antibiotic doses used.

Dose selection has been addressed in Lines 245 to 249.

L225: Technically, this appears to be an intra-abdominal infection model (though there is indeed secondary bacteremia).

We thank the reviewer for the comment on intra-abdominal administration to establish bacteremia. We feel the use of this route to induce bacteremia is in accordance with previously established models and is supported by Supplemental Figure 3 and described in Lines 260 to 261.

L335: What is the rationale for Supplemental Figure 2? The point seems to be the sterility of spleen post-treatment, but there is no parallel 10-day control shown (only 1-hour control is shown).

We appreciate the reviewer's comments and have removed the data point in question and the referenced section within the manuscript. We have retained the remaining data points to show the rapid spread of bacteria to multiple organ sites prior to treatment as mentioned above. *Note this is now Supplemental Figure 3.

REVIEWERS' COMMENTS:

Reviewer #1 (Remarks to the Author):

This manuscript is now much more focused and clear. My queries were rightly considered by the authors.

Please, correct in the final version the term: Gram positive antibiotics; of course antibiotics are not Gram-positive or negative. Should be "Gram-positive" antibiotics.

Reviewer #2 (Remarks to the Author):

-Please indicate the dosing rationale for the bacteremia model. Mice have higher clearance than humans, yet dosing was daily or less.

-“Circumventing resistance to colistin combination therapy” – this is not accurate, as resistance to specific drugs has occurred thus not circumvented. Rather, the authors are presenting alternative combinations. Durability of these secondary combinations have not been tested.

-MCR-1 is closest to EptA of *Moraxella* spp., not LptA of *N. meningitidis*.

Reviewers' comments:

Reviewer #1 (Remarks to the Author):

This manuscript is now much more focused and clear. My queries were rightly considered by the authors.

We appreciate the comment by the reviewer as well as their suggestions throughout the review process.

Please, correct in the final version the term: Gram positive antibiotics; of course antibiotics are not Gram-positive or negative. Should be "Gram-positive" antibiotics.

We thank the reviewer for noting this oversight and have made corrections throughout the manuscript.

Reviewer #2 (Remarks to the Author):

Please indicate the dosing rationale for the bacteremia model. Mice have higher clearance than humans, yet dosing was daily or less.

We thank the reviewer for their comment and have attempted to further clarify the dose selection. Colistin dosing at 7.5 mg/kg was initially selected for the thigh infection model based on dosing used in a similar model by Lui et. al. 2015. Due to the reduction in CFU observed in the thigh model as well as the advantages of dose sparing provided in lines 249 to 251, the colistin dose of 7.5 mg/kg was maintained for the bacteremia model. This has now been outlined in lines 436 to 438.

“Circumventing resistance to colistin combination therapy” – this is not accurate, as resistance to specific drugs has occurred thus not circumvented. Rather, the authors are presenting alternative combinations. Durability of these secondary combinations have not been tested.

Thank you for the comment. We have replaced the term “circumventing” on Line 255.

MCR-1 is closest to EptA of Moraxella spp., not LptA of N. meningitidis.

Thank you for the comment, Lines 360 to 361 have been adjusted accordingly.